# Research on a New Method of Track Turnout Identification Based on Improved Yolov5s

Renxing Chen [1] , Jintao Lv [2], Haotian Tian [2], Zhensen Li [2], Xuan Liu [2] and Yongjun Xie [2,3,*]

1    School of Intelligent Systems Science and Engineering, Jinan University, Zhuhai 510632, China; chenrenxing20@stu2020.jnu.edu.cn
2    International Energy College, Jinan University, Zhuhai 510632, China
3    Institute of Rail Transportation, Jinan University, Zhuhai 510632, China
*    Correspondence: xieyongjun@jnu.edu.cn; Tel.: +86-159-1920-0087

**Abstract:** The modern tram track automatic cleaning car is a crucial equipment in urban rail transportation systems, effectively removing trash, dust, and other debris from the slotted tracks of trams. However, due to the complex and variable structure of turnouts, the cleaning car often requires assistance in accurately detecting their positions. Consequently, the cleaning car needs help in adequately cleaning or bypassing turnouts, which adversely affects cleaning effectiveness and track maintenance quality. This paper presents a novel method for tracking turnout identification called PBE-YOLO based on the improved yolov5s framework. The algorithm enhances yolov5s by optimizing the lightweight backbone network, improving feature fusion methods, and optimizing the regression loss function. The proposed method is trained using a dataset of track turnouts collected through field shots on modern tram lines. Comparative experiments are conducted to analyze the performance of the improved lightweight backbone network, as well as performance comparisons and ablation experiments for the new turnout identification method. Experimental results demonstrate that the proposed PBE-YOLO method achieves a 52.71% reduction in model parameters, a 4.60% increase in mAP@0.5(%), and a 3.27% improvement in precision compared to traditional yolov5s. By improving the track turnout identification method, this paper enables the automatic cleaning car to identify turnouts' positions accurately. This enhancement leads to several benefits, including increased automation levels, improved cleaning efficiency and quality, reduced reliance on manual intervention, and mitigation of collision risks between the cleaning car and turnouts.

**Keywords:** track turnout detection; yolov5s; PP-LCNet; BiFPN; EIoU

## 1. Introduction

The grooves of modern trams are prone to accumulating various types of garbage, including leaves, mud, sand, and debris, which pose a safety risk to train operations and passenger comfort. Therefore, cleaning the slotted tracks is crucial for maintaining the regular operation of trams. In the early stage, the project team developed an automatic track-cleaning vehicle capable of effectively cleaning the slotted tracks of modern trams [1]. However, the cleaning tool of the automated vehicle is susceptible to damage when passing through turnouts due to difficulties in identifying them. Hence, it is necessary to research a highly accurate track turnout identification algorithm, which plays a vital role in achieving a more automated and efficient track-cleaning process [2].

In recent years, extensive research has been conducted both domestically and internationally to address the turnout recognition problem. Traditionally, machine vision methods have been employed for turnout detection. However, these methods have drawbacks such as poor robustness, low efficiency, and susceptibility to environmental effects. The widespread application of deep learning in computer vision fields, including image classification, target detection, and image segmentation [3], has provided a new approach to

turnout detection. Deep learning can handle nonlinear relationships in data, identify and classify data, and possesses strong generalization ability and scalability [4]. Several scholars have attempted to solve the turnout identification problem using deep learning methods. For instance, OLGA et al. [5] utilized convolutional neural networks to automatically extract railroad features. Wenqi Liu [6] employed a deep neural network-based learning algorithm to recognize and predict railroad scene images. The training dataset for this algorithm consisted of railroad scene images, and accurate recognition and prediction of such images were achieved by training the deep neural network model through multiple iterations. Yilmazer M [7] et al. proposed a new method based on yolov4, using the darknet53 backbone of the yolov4 network to train railroad data collected with autonomous drones. They tested the yolov4 network with the darknet53 backbone and applied the model to track turnout scene recognition. He Sen et al. [8] utilized a line array industrial camera scan to obtain railroad point cloud information. They designed a residual connected railroad turnout scene recognition network, employed a tree structure Parzen estimation algorithm to search for optimal hyperparameters, and utilized a focal loss function to address the problem of an imbalanced number of samples, achieving accurate and rapid recognition of railroad turnout scenes. However, this algorithm needs to be retrained for numerous types of turnouts, necessitating further improvement in its robustness.

In previous studies of yolov5s, the computational limitations of practical application platforms have often been overlooked in favor of improving the network model's detection accuracy. This paper presents an enhanced track turnout detection model called PBE-YOLO, which is based on yolov5. Firstly, PBE-YOLO replaces the yolov5 backbone network with PP-LCNet, reducing the number of parameters in the original model. Secondly, BiFPN technology is incorporated into the head of the baseline model, facilitating the effective fusion of multiscale features to address the accuracy loss caused by the lightweight backbone network. Lastly, this paper adopts EIoU as a replacement for the original GIoU loss function in yolov5, aiming to improve the convergence speed of the network. Experimental results demonstrate that the improved yolov5s model exhibits favorable recognition performance on the collected track turnout dataset.

## 2. Materials and Methods

The methodology for track switch recognition, based on the improved yolov5s architecture, is depicted in Figure 1. The dataset utilized in this study comprises images captured by cameras installed on cleaning vehicles. The overall workflow is as follows: Firstly, real-time captured images undergo preprocessing, which includes dataset partitioning and image enhancement operations. These preprocessing steps serve as inputs for the PBE-YOLO model. Subsequently, yolov5s is improved through the following steps:

- Replacing the yolov5s backbone with PP-LCNet to reduce the parameter count and achieve lightweight improvement;
- Employing BiFPN for feature fusion, integrating features from different spatial resolutions, and addressing the accuracy loss associated with the lightweight model;
- Utilizing the EIoU loss function instead of the original regression loss function to address the problem of mismatched predicted and ground-truth bounding boxes and improve convergence speed.

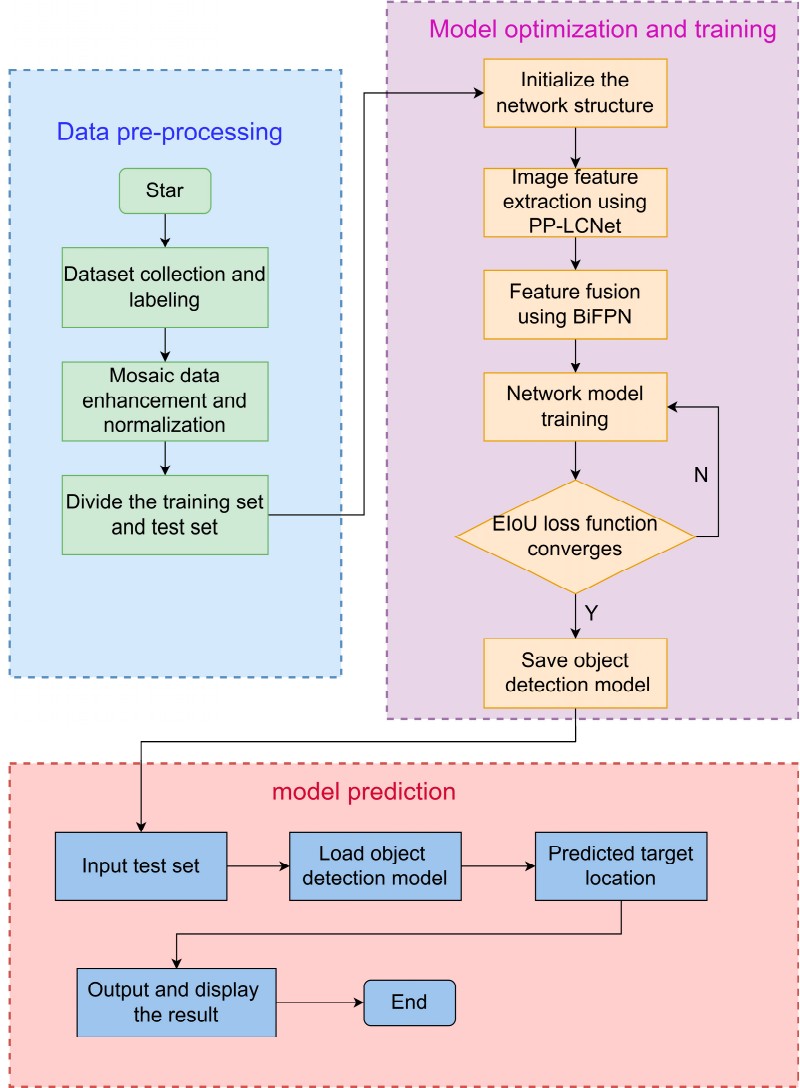

**Figure 1.** Improving the overall flow chart of target recognition of yolov5s.

After initializing the network parameters, the model undergoes training using an optimized regression loss function to obtain the optimal configuration. This training process enables the accurate recognition of track switches by the model. Finally, a validation set is utilized to evaluate the trained model's effectiveness and generate track switch recognition results. This process assesses the model's performance on unseen data, providing an evaluation of its recognition accuracy.

### 2.1. Yolov5s Structure Analysis

Yolov5 is a real-time object detection deep learning model that includes four variants: yolov5s, yolov5m, yolov5l, and yolov5x, representing different model sizes: small, medium, large, and extra-large, respectively. The yolov5 architecture consists of three main components: backbone, neck, and head. The backbone is responsible for feature extraction, while the neck and head handle object detection. Figure 2 illustrates the network architecture of yolov5s, and the following explanations focus on the three crucial components in yolov5 using yolov5s as an example.

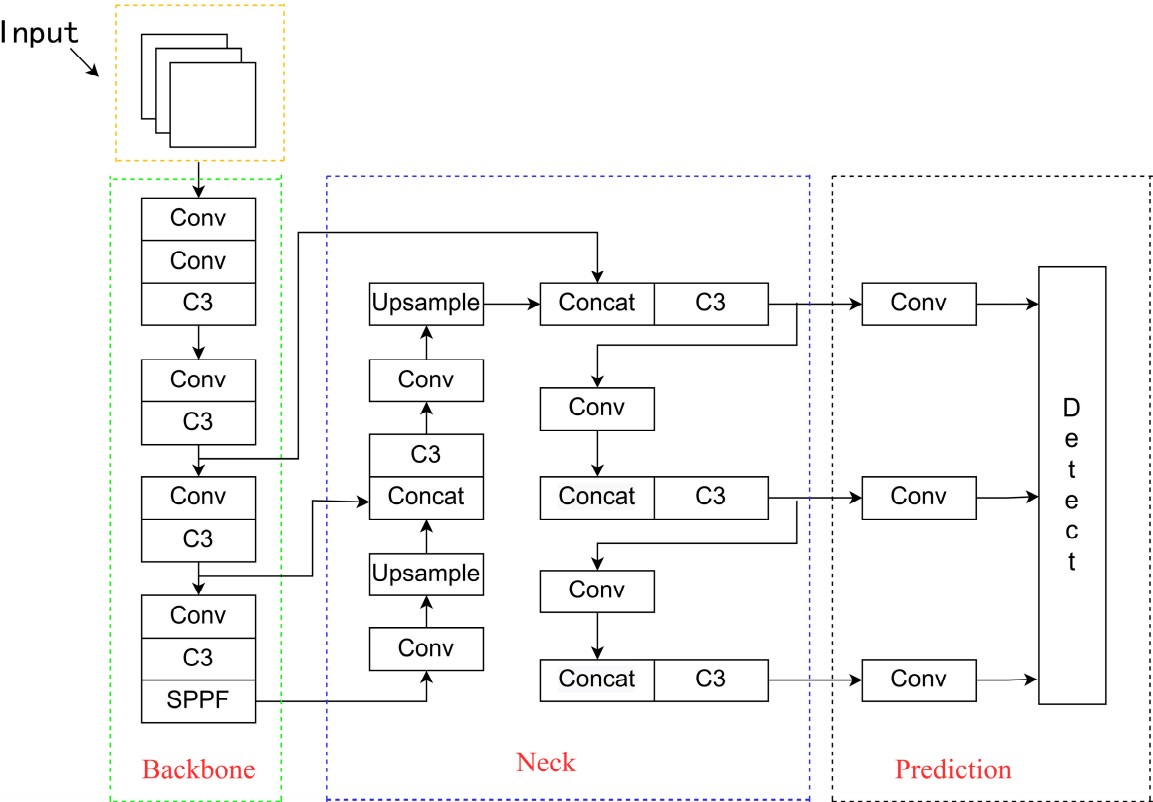

**Figure 2.** Schematic diagram of yolov5s structure.

Backbone: The backbone module primarily consists of the CSP structure [9] and the SPPF module [10]. CSP incorporates convolutional layers, residual blocks, and max-pooling layers, effectively extracting features and maintaining gradient flow. Moreover, the cross-stage connections in CSP increase the depth and width of the network, enhancing its performance. At the end of the backbone, the SPP network [11] captures features of various scales using max-pooling operations with multiple sizes. These features are then fed into the yolov5s detection head to detect objects of different sizes.

Neck: The neck structure resides between the backbone and head and adopts the PAN algorithm [12]. The PAN algorithm consists of convolutional and upsampling layers, aiming to increase feature resolution and fuse features of different scales. Additionally, PAN combines backbone features of different scales to enhance feature representation capabilities further.

Head: The head component consists of a sequence of convolutional layers, with the final prediction layer responsible for object detection. This prediction layer, which is a convolutional layer, predicts the class probabilities, bounding box coordinates, and object scores for each anchor box. The head utilizes anchor-based object detection, where each anchor box detects a specific object. Moreover, the model utilizes the GIoU Loss [13] to estimate the recognition loss associated with the detected bounding boxes.

### 2.2. Yolov5s Backbone Network Lightweight Improvements

PP-LCNet [14] is an efficient convolutional neural network designed to be lightweight. It leverages local connection blocks (LCBlocks) to construct a deep neural network architecture. Figure 3 illustrates the LCBlock module comprising two LocalConv layers and a shortcut connection. Depthwise separable convolutions are utilized within these modules to reduce model parameters and computational complexity, thereby enhancing the efficiency and speed of the network. Additionally, PP-LCNet integrates a lightweight CPU network with MKLDNN acceleration strategy to improve its running speed and efficiency further.

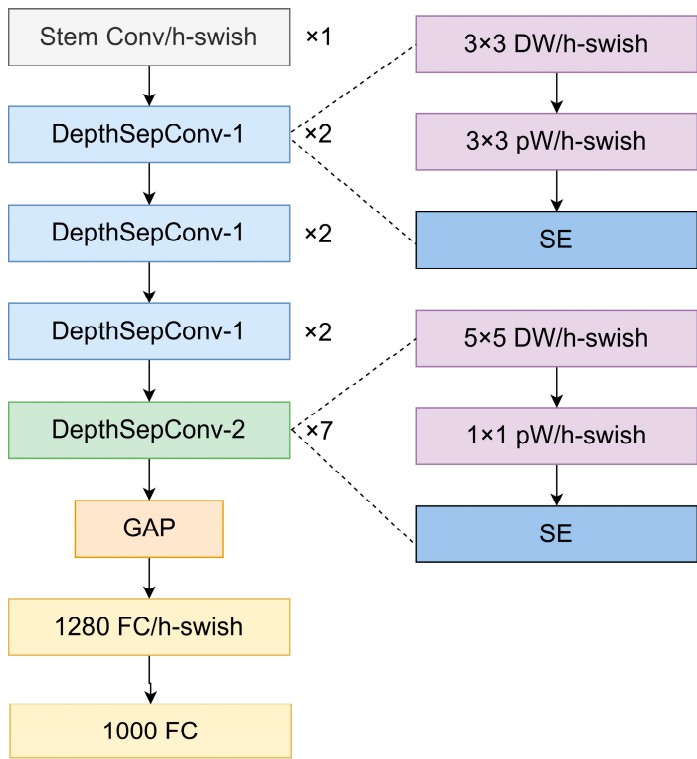

**Figure 3.** PP-LCNet structure diagram.

Traditional convolutional neural networks often improve object detection performance by increasing network depth, channel numbers, and image resolution [15]. While these methods can enhance the network's expressive power, they also significantly increase the computational burden, making model training and inference more challenging. In yolov5, the backbone part adopts the CSP module, which can reduce computational complexity and improve feature representation capabilities [16]. However, this structure is relatively complex and needs to perform better in detecting large-scale objects. Additionally, the lower resolution of the feature maps in CSP may fail to see small objects. On the other hand, PP-LCNet utilizes local connection blocks to construct an efficient deep neural network and employs depthwise separable convolutions to reduce computational complexity further and improve model efficiency. Therefore, in this paper, we consider using PP-LCNet as a replacement for CSPNet in yolov5, referred to as yolov5s_PP-LCNet, in the following sections.

As shown in Figure 4, the improved backbone structure can be divided into nine parts (B1 to B9). Among them, B1 is a module that uses $3 \times 3$ ordinary convolutions for feature extraction, primarily focused on extracting low-level features from the input image. B2 to B8 consists of depth separable convolutions (DepthSepConv) to reduce model parameters and network computation, thereby enhancing computational efficiency. The DepthSepConv mainly consists of structures such as batch normalization (BN), pointwise convolution (PW), and squeeze-and-excitation (SE) blocks. B2 to B7, along with the first convolutional layer of B7, adopt a convolution kernel size of $3 \times 3$, while the subsequent five layers of convolution in B5 and all convolutional layers in B7 use a kernel size of $5 \times 5$. Batch normalization and activation functions are applied to the modules B2 to B7 to increase the non-linear mapping capability of the network and enable it to capture complex patterns in the data. The B8 module introduces the SE block to enhance salient features and suppress unimportant ones, thereby improving its discriminative power. The B9 module corresponds to the SPPF (spatial pyramid pooling fusion) layer in the original backbone, which performs pooling operations of four different sizes ($1 \times 1$, $2 \times 2$, $3 \times 3$, and $6 \times 6$) on the input feature map and concatenates the pooled feature maps together. Based on

this, the SPPF layer can perceive objects at different scales and retain high-level semantic information from the feature maps. Compared to CSPNet, the PP-LCNet structure is more straightforward, easier to train, and performs better in detecting large-scale objects.

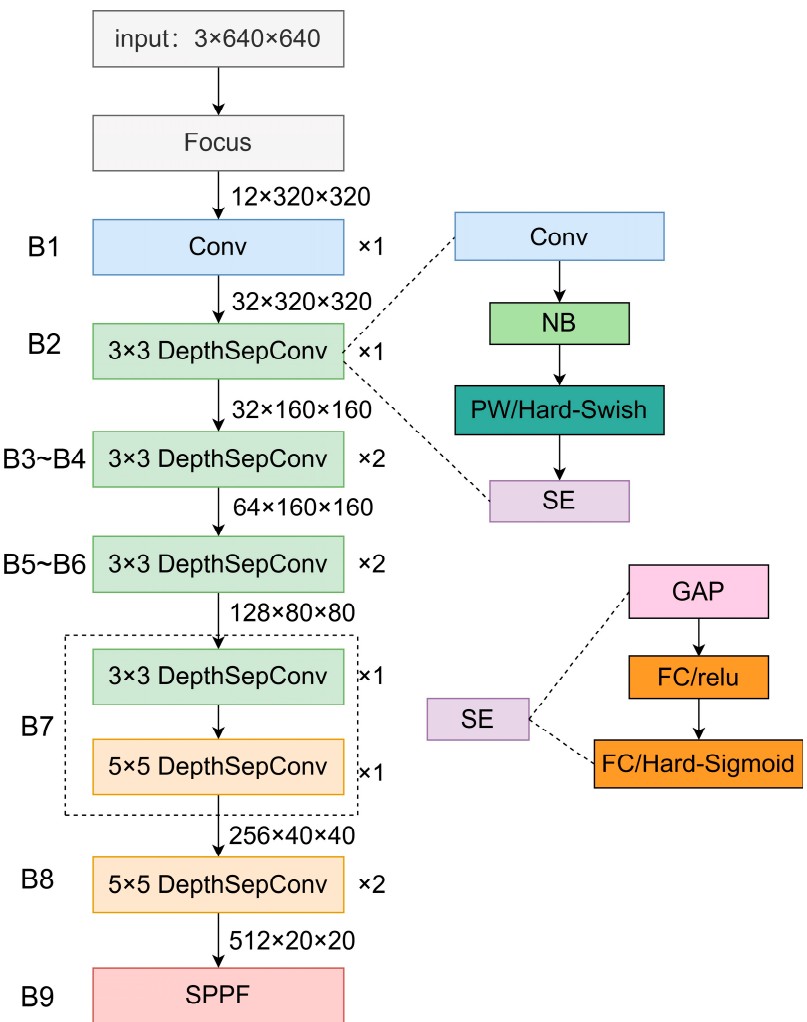

**Figure 4.** Improving the backbone structure diagram in yolov5s.

### 2.3. Improvement of Yolov5s Feature Fusion Method

In the original yolov5s, two feature pyramid networks, PAN (path aggregation network), and FPN (feature pyramid network), are used for feature fusion across layers to integrate features from different scales. As shown in Figure 5A), FPN integrates feature maps of varying resolutions by establishing connections from the top to the bottom of the network. This allows the fusion of high-resolution feature maps from the bottom with low-resolution feature maps from the top, ultimately enhancing the detection accuracy for small objects. As depicted in Figure 5B), PAN adopts a bottom-up and top-down feature fusion approach to restore low-resolution feature maps to the original resolution, but it requires more computational resources. In the case of railway switch detection, the small size of the targets necessitates an improvement in the detection accuracy of the original model. Feature pyramid networks, while beneficial for object detection, may fail to capture fine-grained details, decreasing detection performance.

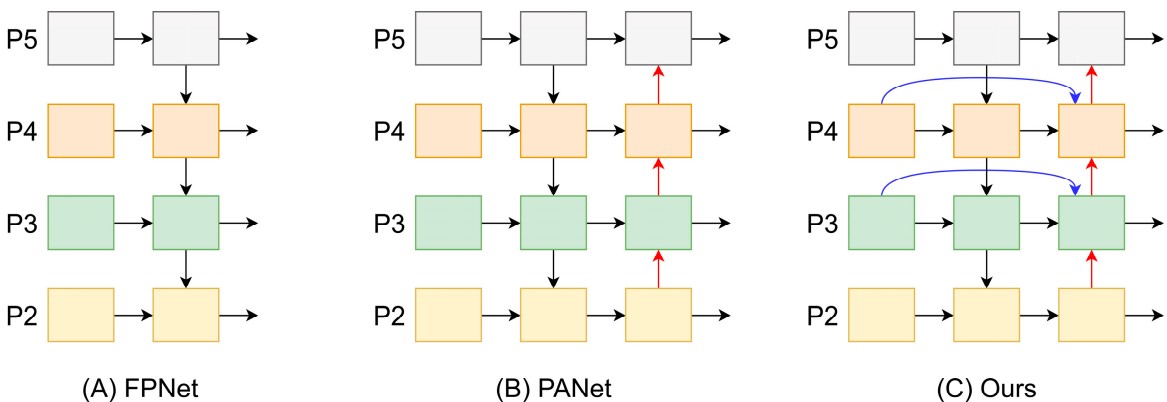

**Figure 5.** Feature Fusion Network Design.

BiFPN (bidirectional feature pyramid network) [17] is an effective method for multi-scale feature fusion that combines feature pyramid networks with bidirectional connections. This fusion technique aims to enhance both detection accuracy and efficiency. Feature pyramid networks enhance object detection accuracy by fusing features of different scales [18]. At the same time, bidirectional connections consider information from both high-level and low-level features, capturing target-specific characteristics more effectively. In BiFPN, each input feature undergoes two convolution operations and a weighted operation separately to extract more useful information from different layers and effectively fuse features of different scales. Furthermore, BiFPN introduces a dynamic weight assignment mechanism that dynamically adjusts the weights between features based on their performance, better-utilizing information from different features. During multi-layer bidirectional connections, BiFPN effectively utilizes information from different levels and facilitates information exchange between different scales [19].

In order to strike a balance between the accuracy loss caused by the lightweight backbone network and the need to detect small objects effectively, this paper improves the original feature fusion method in yolov5s by introducing BiFPN. Specifically, it treats each feature map as a node and represents their relationships through a bidirectional directed graph. In the graph, each node is connected to its contextual nodes, and the weights between each node are learned. For each feature node $i$, the result of weighted fusion can be represented as:

$$y_i = \sum_{j=1}^{n} w_{ij} x_j \tag{1}$$

where $x_j$ denotes the other feature nodes connected to node $i$, $n$ represents the number of nodes connected to node $i$, and $w_{ij}$ indicates the weight between node $i$ and node $j$. BiFPN adopts a learnable weight strategy based on the attention mechanism to enable effective learning and updating of weights between nodes. Each node $i$ is considered an attention head, and a weight $w_{ij}$ is calculated based on the feature map of its context node $j$ and implemented by a neural network with an activation function. The above method allows the weights between nodes to be dynamically adjusted according to the correlation and information between contextual features, resulting in more accurate and robust feature fusion results. The core idea of BiFPN is to introduce cross-scale connections to fuse more feature information while maintaining the exact computational cost. In this paper, the feature layers of the original yolov5s network were improved. As shown in Figure 5B), the original network only fused the feature layers from the 3rd to the 5th layer, neglecting the shallow semantic information from the 2nd layer, which is crucial for detecting small objects. Based on this, we retained the feature fusion layers of the 2nd and 5th layers and added cross-scale connections between the 3rd and 4th layers to obtain the improved architecture, as shown in Figure 5C). The size of these feature maps is typically enlarged to 160 × 160 to enhance object detection accuracy. By introducing the BiFPN layer and utilizing cross-scale connections, the feature maps at different scales are fused and

enhanced, significantly improving object detection performance, particularly in detecting small objects with higher precision.

Building upon the enhancements to the backbone and head outlined in Sections 2.2 and 2.3, the improved yolov5s model is derived, featuring the specific structure illustrated in Figure 6.

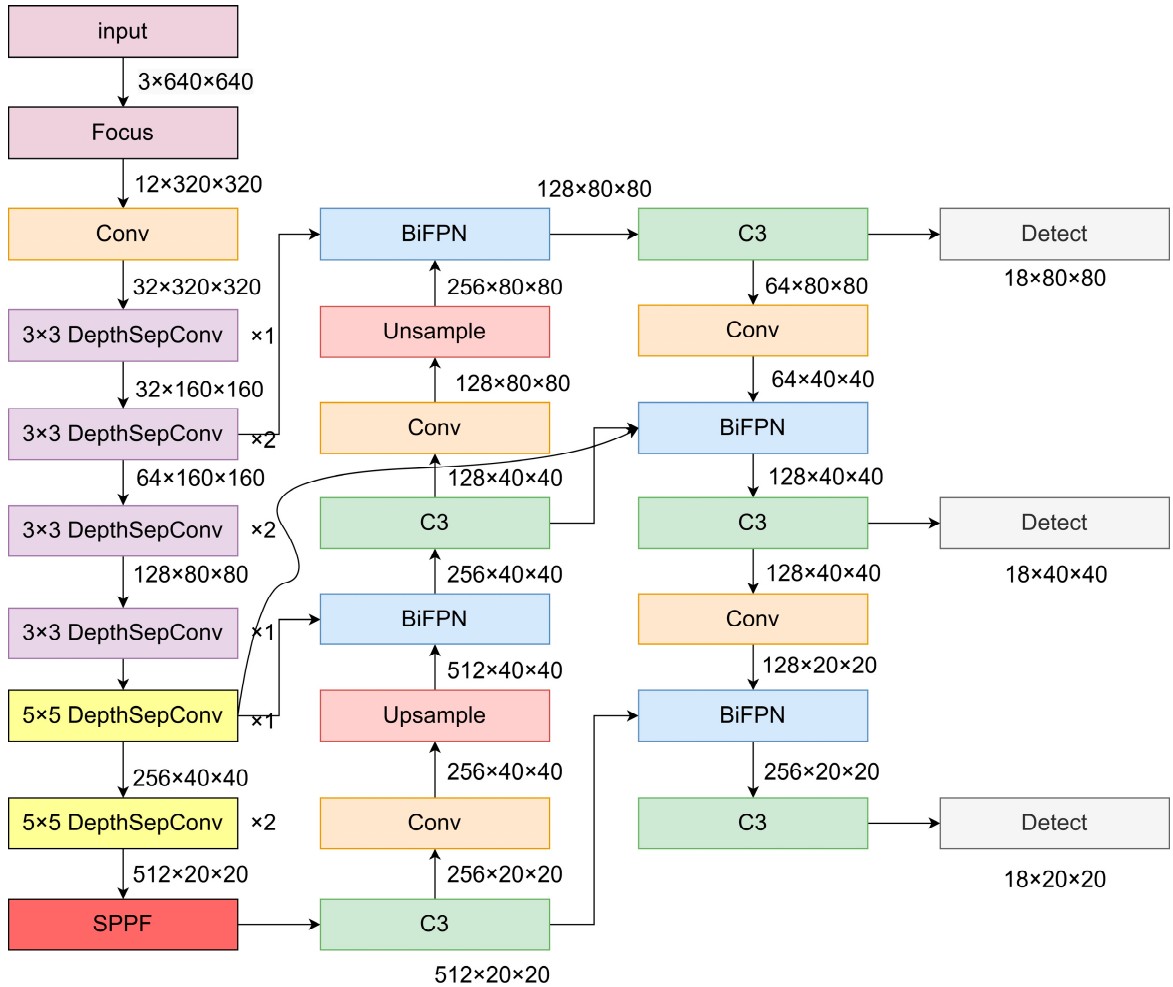

**Figure 6.** Improved yolov5s structure diagram.

### 2.4. Yolov5s Regression Loss Function Optimization

The original yolov5s framework employs the GIoU (generalized intersection over union) loss function, which includes a penalty term compared to the IOU loss function. The GIoU method evaluates the distance between the predicted and ground truth boxes by considering their minimum enclosing rectangles. This evaluation incorporates factors such as the distance between the center points, the variations in width and height, and the area of the enclosing rectangle [20]. However, GIoU does not consider the aspect ratio of the target box. Furthermore, GIoU is sensitive to variations in the target scale. Any changes in scale affect the distances between center points, differences in width and height, and the area of the bounding rectangle, consequently impacting the GIoU calculation results. This paper replaces the GIoU loss function with the EIoU (enhanced intersection over union) loss function to address this issue.

Compared to GIoU, EIoU provides a more accurate measurement of the matching degree between the predicted bounding box and the ground truth bounding box, especially in cases where there is a complex overlap between objects. EIoU can better distinguish the quality of predicted bounding boxes. Additionally, EIoU is more sensitive to the aspect ratio of the bounding box when measuring the matching degree, allowing for better adaptation

to bounding boxes with different aspect ratios. Moreover, when calculating the loss for false positive detections, EIoU sets the IoU metric between the bounding boxes to 0 and considers both position offset and size offset in the loss calculation. This approach enables better differentiation between false detection and missed detection, imposing stricter penalties for false positive detections and improving the accuracy of bounding box regression. The definition of EIoU is as follows:

$$
\begin{aligned}
L_{EIoU} &= L_{IoU} + L_{dis} + L_{asp} \\
&= 1 - IoU + \frac{\rho^2\left(b,b^{gt}\right)}{c^2} + \frac{\rho^2\left(w,w^{gt}\right)}{C_w^2} + \frac{\rho^2\left(w,w^{gt}\right)}{C_h^2}
\end{aligned}
\tag{2}
$$

where $c_w$, $c_h$, and $\rho$ are the width, height, and Euclidean distance between $b$ and $b^{gt}$ of the minimum external box covering the two boxes, respectively, and $w, h, w^{gt}, h^{gt}$ are the box heights of the predicted and actual boxes, respectively.

### 3. Results and Discussion

#### 3.1. Datasets and Experimental Platforms

The dataset plays a crucial role in object detection tasks. Factors such as the dataset's size, quality, and diversity directly impact the model's training effectiveness and generalization capability. However, there is a limited availability of comprehensive datasets specifically for railway turnout detection. This paper uses a dataset of railway turnouts collected from real-world tramway lines for research purposes. The final dataset consists of 1122 images of railway turnouts along with corresponding annotations in the txt format. The dataset is divided into a training set and a testing set in an 8:2 ratio. There is only one recognition object (category) in our dataset, which is the track switch. Furthermore, cleaning vehicles is usually performed during the daytime and under favorable weather conditions. Therefore, our dataset does not consider adverse conditions such as nighttime or rainy weather. Figure 7 displays three sample images from the railway turnout dataset. Before training, all data undergo preprocessing steps such as Mosaic data augmentation and normalization.

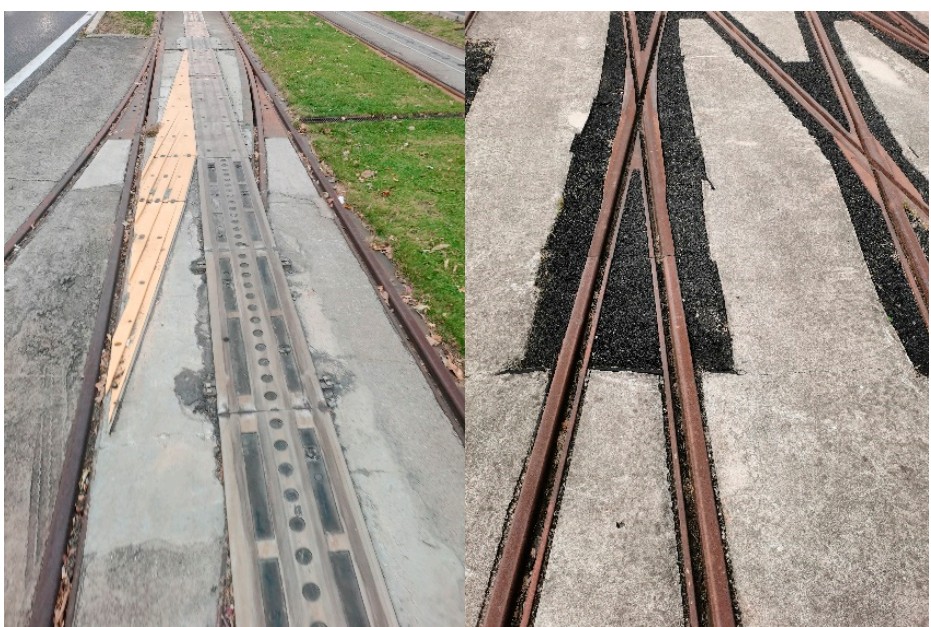

**Figure 7.** Example of dataset images.

#### 3.2. Experimental Data Processing

In yolov5, data augmentation refers to a series of transformations and expansions applied to the training data to increase its diversity and richness. By employing data

augmentation techniques such as random transformations like rotation, scaling, translation, flipping, etc., the model can learn the invariances and variabilities of different objects while also expanding the size of the dataset.

In addition to the fundamental data augmentation techniques, yolov5 incorporates the Mosaic data augmentation method. The core concept behind Mosaic data augmentation is to combine multiple training images into a single composite training image while simultaneously adjusting the positions and sizes of the bounding boxes. This unique image is then used as input for training the object detection model. An example of the Mosaic data augmented image is shown in Figure 8. Precisely, the Mosaic data augmentation method consists of the following steps:

1. Randomly select four different training images.
2. Concatenate these four images together in a specific order to form a new training image. Typically, the four images are divided into two rows, with the left two images forming the top half and the right two images forming the bottom half.
3. Calculate the width and height of the composite image.
4. Adjust the bounding boxes within the new image. For each bounding box, convert its coordinates to be relative to the top-left corner of the new image and scale them according to the scaling factor of the new image.
5. Apply other data augmentation operations, such as random scaling, translation, flipping, etc.

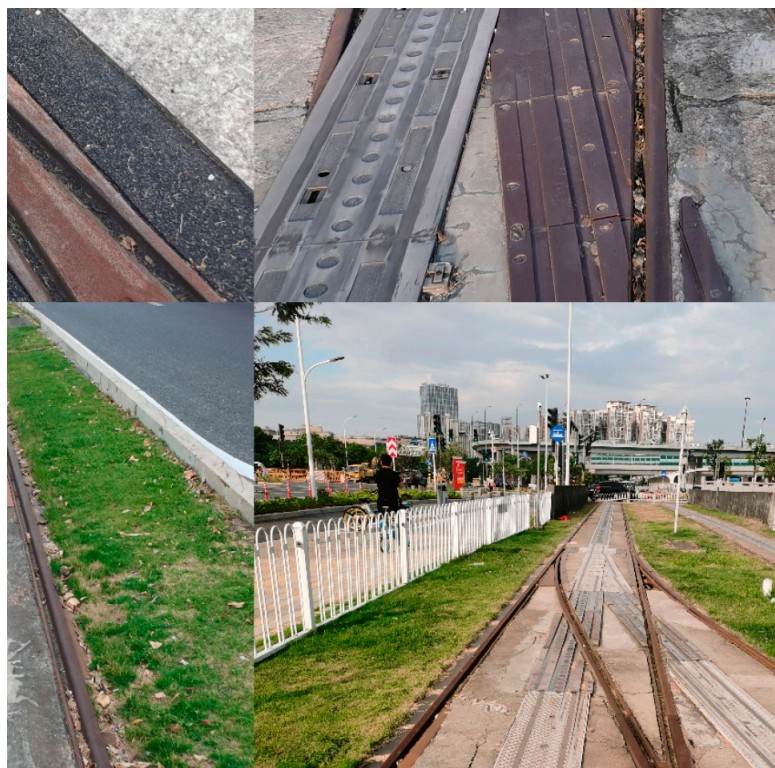

**Figure 8.** Mosaic data augmented image.

Using the Mosaic data augmentation method, the PBE-YOLO model can learn multiple scenes and objects simultaneously within a single image, thereby increasing the diversity and complexity of the data. This helps improve the model's robustness and generalization, enabling it to better adapt to various scenes and object-detection tasks. Additionally, since the Mosaic data augmented image is composed of four original images, there is a higher probability of including small objects in each image, which benefits the performance of small object detection.

*3.3. Experimental Environment and Evaluation Index*

The hardware platform used in the experiments includes:

(1)　CPU: Intel(R) Xeon(R) Platinum 8358 CPU @ 2.60 GHz
(2)　GPU: NVIDIA A800 PCIe

The workstation uses CUDA 11.4 and cuDNN 11.4 to leverage GPU acceleration. The chosen deep learning framework is PyTorch, and the operating system is Linux. The experiments employ stochastic gradient descent (SGD) to optimize the learning rate during training. The hyperparameters are configured as follows: weight decay = 0.0005, momentum = 0.8, batch size = 64, and the number of epochs = 300.

To evaluate the effectiveness of PBE-YOLO, the model's performance is assessed by comparing the detection results between the original yolov5s and PBE-YOLO. The selected evaluation metrics include recall rate (R), precision (P), mean average precision (mAP), parameter count, GFLOPs, and frames per second (FPS). FPS represents the number of images the model can process per second and is typically used to measure real-time performance.

The calculation formulas for recall rate, precision, and average precision are as follows:

$$R = \frac{TP}{TP + FN} \tag{3}$$

$$P = \frac{TP}{TP + FP} \tag{4}$$

$$AP = \int_0^1 P(R)dR \tag{5}$$

In the formulas, TP denotes the count of predicted bounding boxes with an IoU greater than 0.5, belonging to the same class as the ground truth boxes. FP represents the count of predicted bounding boxes with an IoU less than or equal to 0.5 or an IoU greater than 0.5 but not belonging to the same class as the ground truth boxes. FN denotes the count of ground truth boxes that are not predicted.

*3.4. Loss Function Comparison*

Figure 9 illustrates the loss variation during the training process of the original yolov5 and the improved PBE-YOLO model. It is noticeable that the loss value of the PBE-YOLO model shows a rapid initial decrease with more significant fluctuations. However, as the number of training epochs increases, the fluctuations gradually decrease, particularly after approximately 100 epochs. The loss curve of EIoU gradually decreases and stabilizes. Around 250 epochs, the algorithm's loss becomes stable, indicating convergence of the model. Moreover, the loss value of PBE-YOLO is relatively lower compared to GIoU, indicating better robustness of the PBE-YOLO model.

*3.5. Comparative Experimental Analysis of the Performance of Improved Lightweight Networks*

To verify the effectiveness of the improved lightweight backbone network in yolov5s_PP-LCNet, comparative tests were conducted with popular lightweight backbone networks currently in use, namely EfficientLite [15], MobileNetv3 [21], and ShuffleV2 [22]. Evaluation metrics such as precision (P), recall (R), mean average precision at IoU 0.5 (mAP@0.5), number of parameters (parameters), computational complexity (GFLOPs), and frames per second (FPS) were chosen. The comparative test results are summarized in Table 1.

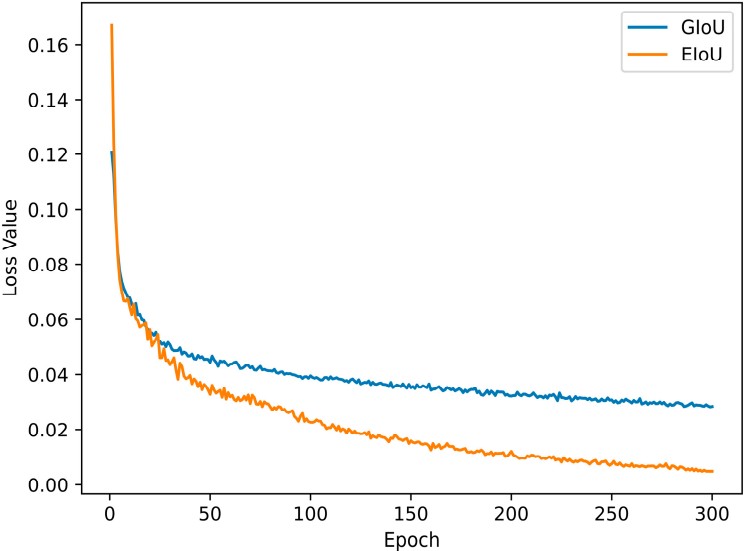

**Figure 9.** Comparison of loss results.

**Table 1.** Comparison results of lightweight models.

| Method | P (%) | R (%) | mAP@0.5 (%) | Params (M) | GFLOPs | FPS |
|---|---|---|---|---|---|---|
| yolov5s(baseline) | 69.15 | 63.43 | 57.42 | 7.02 | 15.9 | 93.46 |
| yolov5s_EfficientLite | 67.24 | 58.35 | 53.01 | 3.77 | 7.4 | 78.74 |
| yolov5s_MobileNetv3 | 67.28 | 59.16 | 53.64 | 3.53 | 6.1 | 81.30 |
| yolov5s_Shufflev2 | 66.68 | 57.50 | 53.21 | 3.18 | 5.9 | 111.11 |
| yolov5s_PP-LCNet(improved) | 68.90 | 63.83 | 55.07 | 3.32 | 6.2 | 99.01 |

Table 1 illustrates that the improved lightweight network, yolov5s_PP-LCNet, outperforms other lightweight networks regarding P, R, and mAP@0.5. Compared to yolov5s_Shufflev2, yolov5s_PP-LCNet achieves a 3.30% improvement in P and an 11.01% improvement in R. Furthermore, yolov5s_PP-LCNet exhibits advantages in terms of the number of parameters, GFLOPs, and FPS compared to yolov5s. The number of parameters is reduced by 53.8%, mAP@0.5 is reduced by approximately 4.10%, and FPS is increased by 5.94%. Therefore, yolov5s_PP-LCNet demonstrates better computational performance, improving network efficiency and reducing the computational burden. Although yolov5s_Shufflev2 outperforms yolov5s_PP-LCNet regarding parameters, GFLOPs, and FPS, it suffers from a high accuracy loss of 7.33%. Yolov5s_EfficientLite and yolov5s_MobileNetv3 show lower metrics than the baseline model, with accuracy losses exceeding 6% relative to yolov5s_PP-LCNet, and lower FPS. Considering the comprehensive comparison, the enhanced lightweight network, yolov5s_PP-LCNet, successfully reduces the parameter count and computational complexity, resulting in a lightweight improvement compared to yolov5s. This advancement facilitates the deployment and application of the model on resource-constrained devices.

*3.6. Performance Comparison Experiments of Different Models*

To further validate the effectiveness of the PBE-YOLO method in track switch recognition, a comparative analysis was conducted with several other object detection models, namely yolov5s, yolov3, yolov3_tiny, yolov4, and yolov4_tiny. The experimental results are presented in Table 2.

**Table 2.** Performance comparison results of different models.

| Method | P (%) | R (%) | mAP@0.5 (%) | Params (M) | GFLOPs | FPS |
|---|---|---|---|---|---|---|
| yolov5s(baseline) | 69.15 | 63.43 | 57.42 | 7.02 | 15.9 | 93.46 |
| PBE-YOLO(Ours) | 71.41 | 65.54 | 60.06 | 3.24 | 6.2 | 99.15 |
| Yolov3 | 65.18 | 55.20 | 51.86 | 61.52 | 155.3 | 74.63 |
| Yolov3-tiny | 63.54 | 53.20 | 50.49 | 8.67 | 12.99 | 212.77 |
| Yolov4 | 66.05 | 58.35 | 52.90 | 60.43 | 131.6 | 54.95 |
| Yolov4-tiny | 62.72 | 58.35 | 51.45 | 3.06 | 6.409 | 169.49 |

Table 2 shows that the PBE-YOLO model exhibits higher performance in terms of precision and mAP@0.5, reaching 71.41% and 60.06%, respectively, which are 3.27% and 4.60% higher than the baseline model. Compared to other object detection models, PBE-YOLO also demonstrates superior performance. Particularly in terms of mAP@0.5, PBE-YOLO outperforms yolov3, yolov3_tiny, yolov4, and yolov4_tiny by 15.81%, 18.95%, 13.53%, and 16.73%, respectively.

Regarding recall, the PBE-YOLO model achieves 65.54%, slightly higher than the 63.43% of yolov5s. Compared to other models, it performs better, significantly outperforming yolov3_tiny and yolov3 by 23.20% and 18.73%, respectively.

Regarding model parameters and computational complexity, the PBE-YOLO model has 3.24 million parameters and 6.2 GFLOPs of computational complexity, smaller than yolov5s with 7.02 million parameters and 15.9 GFLOPs. Moreover, compared to other object detection models, the parameter and computational complexity of the PBE-YOLO model are also significantly smaller. Specifically, compared to yolov3 and yolov4, the parameter reduction is 94.73% and 94.64%, respectively, while the computational complexity reduction is 96.01% and 95.29%, respectively. Thus, the improved PBE-YOLO model optimizes model size and parameter count while improving detection accuracy, providing greater flexibility for model deployment on application devices.

Regarding detection speed, PBE-YOLO achieves a frame rate of 99.15, demonstrating its computational efficiency. Its performance surpasses other algorithms, such as yolov3, yolov4, and yolov4_tiny, with frame rates of 74.63, 54.95, and 169.49, respectively. Although yolov3_tiny achieves a peak FPS of 212.77, its performance in terms of P and recall is significantly lower than that of PBE-YOLO. Therefore, PBE-YOLO balances accuracy and computational efficiency, delivering competitive FPS while maintaining high precision.

*3.7. Ablation Experiments*

To better validate the impact of the three improvement modules—lightweight backbone network, feature fusion method, and regression loss function optimization—proposed in PBE-YOLO on the recognition performance, this section will break down the model and conduct ablation experiments by gradually incorporating the improvement modules. After the experiments, the effects of each module on the model will be analyzed, and the changes in accuracy for each model will be compared. Table 3 displays the results of the ablation experiments.

**Table 3.** Comparison of ablation experiment results.

| Methods | P (%) | mAP@0.5 (%) | Params (M) |
|---|---|---|---|
| yolov5s(baseline) | 69.15 | 57.42 | 7.02 |
| yolov5s + PP-LCNet | 68.90 | 55.07 | 3.32 |
| yolov5s + PP-LCNet + BiFPN | 70.89 | 58.15 | 3.24 |
| yolov5s + PP-LCNet + BiFPN + EIoU(PBE-YOLO) | 71.41 | 60.06 | 3.24 |

Table 3 reveals that by replacing the backbone network of yolov5s with the proposed PP-LCNet, the parameter count decreases from 7.02 million to 3.32 million, resulting in a significant reduction of 52.71% compared to the original model. However, the mAP@0.5 metric decreases by 4.09%. Therefore, directly applying the PP-LCNet network results in lightweight characteristics but sacrifices a significant amount of accuracy, making it unsuitable for direct use.

Building upon the lightweight backbone network, improving the structure of the FPN by using BiFPN, compared to the lightweight model, the P increases by 2.89% to 70.89%, and the mAP@0.5 increases by 5.59% to 58.15%. The notable improvement in parameter reduction is attributed to the practical feature map fusion ability of BiFPN, which enables the capture of semantic information from objects at various scales. Through bidirectional propagation, BiFPN can transfer information between different levels, guiding and complementing low-level features with high-level features. Thus, it can improve the model's accuracy while reducing parameter count and computational complexity.

Finally, replacing the original yolov5s' regression loss function GIoU with EIoU significantly impacts detection accuracy. The final P of the model is 71.41%, an increase of 0.73%, and the mAP@0.5 is 60.06%, an increase of 3.28%. Therefore, the improved EIoU enhances the accuracy of the network, making it more sensitive to track switch objects and receiving more attention without increasing the parameter count. Thus, it is more suitable for a lightweight model.

### 3.8. Experimental Analysis of the Application Effect Verification of the New Method of Turnout Identification

Finally, in this study, a validation dataset was collected that did not appear in the model training process. The best models trained using the original yolov5s and PBE-YOLO were used to detect images from this validation dataset, as shown in Figure 10. Through the experiments, it was found that the original yolov5s algorithm already achieved high accuracy during the training process. However, there is still significant room for improvement in performance in real-world detection scenarios. As shown in Figure 10a, it is evident that the baseline model yolov5s has missed detections. On the other hand, in Figure 10b, it is apparent that the improved yolov5s can accurately detect more miniature track switches. In Figure 10c, although the original yolov5s model can detect track switches in the image, the confidence scores for the detected objects are relatively low, ranging from 0.65 to 0.75. In contrast, the improved yolov5s model shows higher confidence scores (Figure 10d), ranging from 0.70 to 0.85, indicating that the improved yolov5s performs better in detecting track switches.

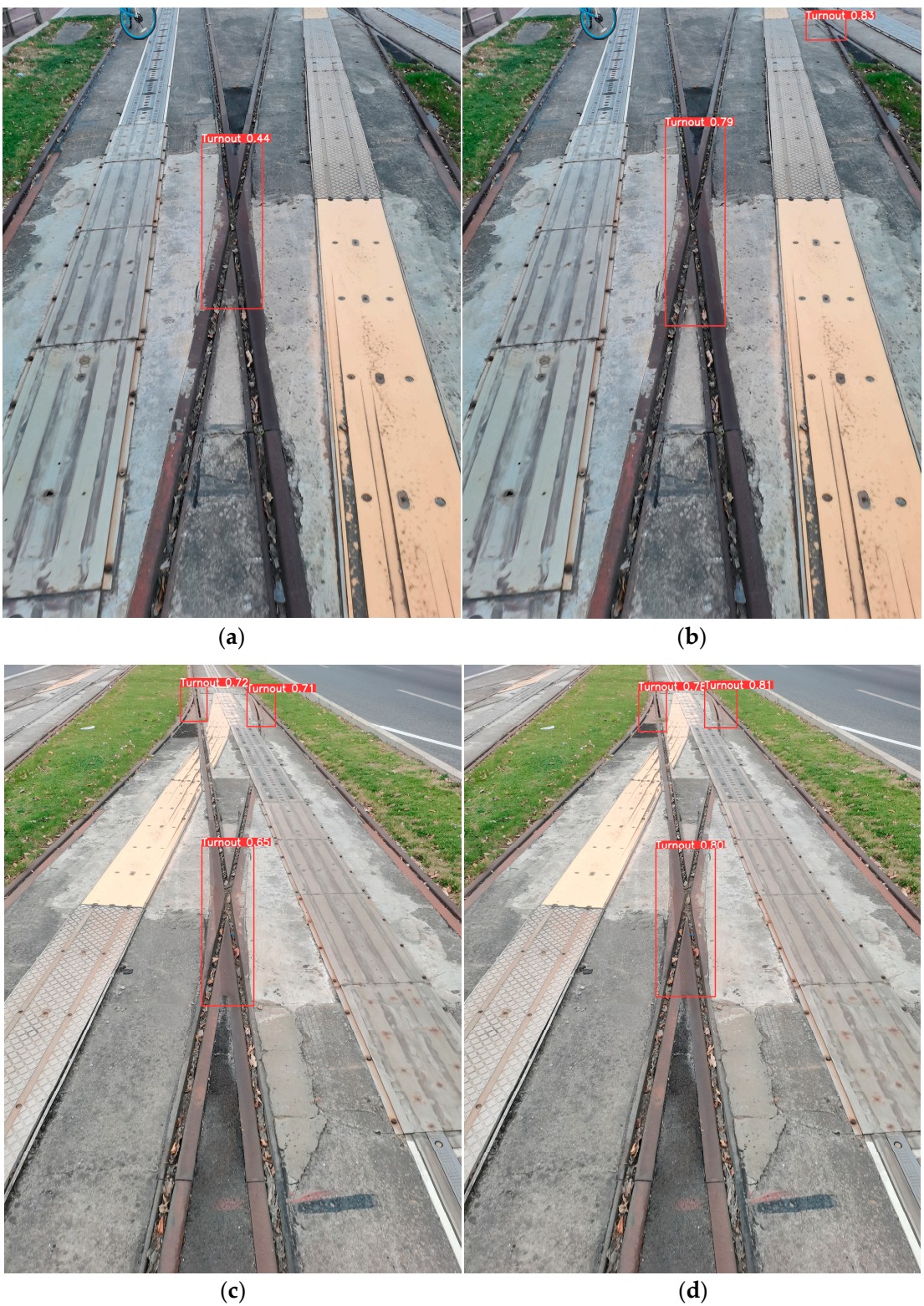

**Figure 10.** Comparison of detection results before and after improvement of yolov5s (**a**,**c**) graphs are yolov5s detection results; (**b**,**d**) graphs are PBE-YOLO detection results.

## 4. Conclusions

This paper proposes a new method called PBE-YOLO for track switch object recognition based on the improved yolov5s architecture. The objective is to address the challenge of accurately identifying the position of track switches for existing track-cleaning vehicles.

The proposed method encompasses several enhancements, namely replacing the backbone with PP-LCNet to reduce parameter count and computational complexity, integrating BiFPN feature fusion to address accuracy loss in the lightweight design, and utilizing the EIoU loss function to improve convergence. These enhancements collectively contribute to the improved performance of the method. Experimental results demonstrate that PBE-YOLO achieves improved detection accuracy while maintaining lightweight characteristics, providing theoretical and technical support for the automatic recognition of track switches. Compared to the original yolov5s and other models, PBE-YOLO demonstrates superior detection accuracy and fewer model parameters when evaluated on the track turnouts dataset. Therefore, the application of PBE-YOLO is expected to play a crucial role in practical track-cleaning vehicle projects, enhancing cleaning efficiency and quality, reducing manual intervention, and mitigating the risk of collisions between cleaning vehicles and track switches. Future research can further explore incorporating attention mechanisms and other operations to improve model accuracy and facilitate its deployment and application on track-cleaning vehicle hardware platforms.

**Author Contributions:** R.C. contributed to the methodology, software development, and improvement of BiFPN feature fusion and EIoU loss function. R.C., J.L. and H.T. contributed to the study's conception and design. Z.L. organized the database. R.C. performed the statistical analysis. Z.L. wrote the initial draft of the manuscript. R.C., H.T., Z.L. and X.L. contributed to writing different manuscript sections. Y.X. is responsible for the revision of the paper. All authors have read and agreed to the published version of the manuscript.

**Funding:** This research was funded by the National Innovation and Entrepreneurship Training Program for Undergraduate (No.:202210559109), with a funding amount of 20,000 yuan.

**Data Availability Statement:** The original contributions presented in this study are included in the article. Further inquiries regarding data availability can be directed to the corresponding author.

**Acknowledgments:** We would like to acknowledge the support received not covered by the author contributions or funding sections. This may include administrative and technical assistance or donation of experiment materials.

**Conflicts of Interest:** The authors declare no relevant conflict of interest.

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
