# Peer review of "Research on a New Method of Track Turnout Identification Based on Improved Yolov5s"

_processes, doi:10.3390/pr11072123_

Round 1
Reviewer 1 Report
“However, it is difficult for the cleaning car to recognize the turnouts automatically when it is at the turnouts.”
The significance of this study is not explicitly stated in the abstract. It is recommended to expand the scope of the research.
“The algorithm improves yolov5s based on the structure of yolov5s, including backbone net-work lightweight improvement, feature fusion improvement and loss function optimization.”
The author's improvements to the original algorithm lack significant innovation in this section. Additionally, there is a lack of explanation regarding the reasons for making three specific modifications in relation to the research subject. It is suggested to provide supplementary information in this regard.
“The experiments show that the proposed new method of track turnout identification, PBE-YOLO, compared with the traditional yolov5s, the number of model parameters decreases by 52.71%, [email protected](%) increases by 4.60%, and the average accuracy increases by 3.27%, which shows that the new method effectively lightens the model and also improves the detection accuracy and precision of the algorithm.”
The experimental results obtained by the author appear to be less satisfactory, with a low mAP (mean average precision) value. It is recommended to consider re-annotating the dataset and conducting further experiments.
The author conducted various model comparisons and ablation experiments in the experimental section. However, the inconsistency between the average precision (AP) and mean average precision (mAP) suggests that there are multiple annotated classes in the experiments. It is recommended for the author to supplement the information about other detection objects in the paper.
Some basic formatting and grammar improvements are needed.
Reviewer 2 Report
The authors presented an article titled: “ Research on a new method of track turnout identification based 2 on improved yolov5s ”. In this paper the authors proposed an improved yolov5s to recognize the track turn.The article is interesting and very useful in rail transit field, however, there are several points in the article that require further explanation.
1.Abstract must be improved. Present in the abstract novelty, practical significance of presented method.
2.The paper lacks the verification of various light and weather conditions (such as night, rainy day, etc.), only a single scene is verified, and the amount of training data is not given by the authorï¼›
3. Please describe in detail how the author does data enhancement. The paper does not give too much description. The quality of data samples and data has a direct impact on the accuracy of the model;
4. The author does not consider the efficiency of the model, especially in the field of transportation. The efficiency of the model (the number of frames processed per second) is a key indicator of the usability of the model.Please provide additional explanation and analysisï¼›
5. The author describes the optimized loss function in the paper, please draw the loss function graph and analyze it;
6. The paper only shows the recognition effect of one scene, which is not convincing. Please add more data for analysis and display. Secondly, the recognition accuracy of the model is not high, so whether it can be really applied is still worth careful consideration.
7.Add quantitative and qualitative work results. In addition, it is necessary to more clearly show the novelty of the article and the advantages of the proposed method. What is the difference from previous work in this area? Show practical relevance. Presented conclusions are only a description of the test results. Conclusions should reflect the purpose of the article.
The article should be proofread by a native English speaker. The authors must carefully study the comments and make improvements to the article step by step. Mark all changes in color.
Reviewer 3 Report
The manuscript provides a clear introduction to the problem addressed in the paper, which focuses on the difficulty of automatic recognition of turnouts by modern tram automatic cleaning vehicles. The proposed method, called PBE-YOLO, utilizes an improved version of the YOLOv5s algorithm for track turnout recognition. The improvements include lightweight enhancements to the backbone network, feature fusion improvements, and optimization of the loss function.
The manuscript mentions that the performance of the improved backbone network is compared through experimental analysis using track turnout datasets collected from modern tram lines. The new method is evaluated through performance and ablation experiments, comparing it with traditional YOLOv5s. The results indicate that PBE-YOLO reduces the number of model parameters by 52.71%, increases [email protected](%) by 4.60%, and improves average accuracy by 3.27%. These findings demonstrate the effectiveness of the proposed method in reducing model complexity while improving detection accuracy and precision.
The manuscript concludes by mentioning the validation of the new method for practical applications using a validation set, indicating that the proposed approach is applicable in real-world scenarios.
Minor comments:
It would be beneficial to include specific details about the field photography dataset, such as the size, diversity, and quality of the collected images. This information would enhance the understanding of the dataset used for training and evaluating the PBE-YOLO method.
It would be helpful to briefly mention the metrics used to assess the performance of the proposed method in the performance and ablation experiments. Providing this information would give readers a better understanding of how the improvements in model parameters, [email protected](%), and average accuracy were measured.
Based on the clarity of the paper and the promising results presented, I recommend accepting this article for publication with the suggested minor comments addressed.
Minor editing of English language required
Reviewer 4 Report
The paper titled "Research on a New Method of Track Turnout Identification based on Improved YOLOv5s" introduces an innovative approach for recognizing track turnouts using YOLOv5s. This method enhances the algorithm's detection accuracy and precision, leading to improved detection results.
However, I have the following questions,
(1) The author argues that traditional methods suffer from drawbacks such as limited robustness, low efficiency, and vulnerability to environmental factors. However, considering Figure 7, is it feasible to utilize traditional computer vision (CV) methods to segment groove lines and examine cross-points for turn-out detection? In case traditional CV methods prove inadequate, it would be beneficial if the authors could provide more challenging scenarios for traditional CV approaches to further support their concept.
(2) I have another important question regarding YOLO. In YOLO, labeling bounding boxes is necessary, typically using rectangles to indicate the object's location. However, for turn-outs, there is no naturally defined contour or fixed aspect ratio for bounding boxes. From Figure 8, it is evident that the bounding boxes have varying rectangle ratios. How were these bounding boxes defined? This is a crucial aspect because, in the case of turn-outs, bounding boxes may not be the most suitable choice. Key point prediction could potentially be a better approach. Additionally, I'm curious about the labeling process for the 1122 turn-outs. Were they all labeled by a single person, or were multiple individuals involved? If multiple people were involved in the labeling process, it could introduce larger variances in the bounding boxes, thereby potentially affecting the algorithm's performance.
(3) I have another question regarding the accuracy of YOLO, which is currently around 0.7. Has the author considered incorporating additional methods to improve the results? For instance, combining the outputs of different methods such as traditional computer vision techniques, YOLOv5, and the Segment Anything Model (SAM) enhances the robustness of their results. This approach could potentially yield more accurate and reliable outcomes.
(4) I have another concern regarding the YOLO model used in their research. I couldn't find any information or link on the specific YOLO model code they utilized. I assume the authors didn't develop their own YOLO model from scratch, as it typically involves complex implementation techniques and extensive training. If they did implement their own YOLO model accurately, it would be beneficial to see their model's performance on well-established datasets like COCO to demonstrate its efficiency and effectiveness.
Some small errors,
(1) In line 90, it shall be “analysis” not “analysi”
(2) I have a question regarding line 228. The term "CIoU" is mentioned, but I wonder if it is a typo for "GIoU"?
Generally good, just a few small typos that need to be fixed.
Round 2
Reviewer 1 Report
The author has made appropriate improvements according to the review, and I have no other suggestions on the current version.
English writing is accetable.
Author Response
Thank you for your valuable comments on the article.
Reviewer 2 Report
The authors have answered all the questions I am concerned. The paper can be accepted in present form
The English writing skill needs to be improved. I found the grammatical errors in some places.
Author Response
We appreciate your valuable comments and suggestions regarding the article.
Reviewer 4 Report
The authors address my concerns and I am good with the paper now.
Author Response
We would like to express our gratitude for your valuable comments and suggestions on the article.